# Design and Evaluation of Capacitive Smart Transducer for a Forestry Crane Gripper

**DOI:** 10.3390/s23052747

**Published:** 2023-03-02

**Authors:** Narendiran Anandan, Dailys Arronde Pérez, Tobias Mitterer, Hubert Zangl

**Affiliations:** Sensors and Actuators Group, Institute for Smart System Technologies, Universität Klagenfurt, 9020 Klagenfurt am Wörthersee, Austria

**Keywords:** grasp detection, capacitive sensing, IEEE 1451.0, forestry gripper

## Abstract

Stable grasps are essential for robots handling objects. This is especially true for “robotized” large industrial machines as heavy and bulky objects that are unintentionally dropped by the machine can lead to substantial damages and pose a significant safety risk. Consequently, adding a proximity and tactile sensing to such large industrial machinery can help to mitigate this problem. In this paper, we present a sensing system for proximity/tactile sensing in gripper claws of a forestry crane. In order to avoid difficulties with respect to the installation of cables (in particular in retrofitting of existing machinery), the sensors are truly wireless and can be powered using energy harvesting, leading to autarkic, i.e., self-contained, sensors. The sensing elements are connected to a measurement system which transmits the measurement data to the crane automation computer via Bluetooth low energy (BLE) compliant to IEEE 1451.0 (TEDs) specification for eased logical system integration. We demonstrate that the sensor system can be fully integrated in the grasper and that it can withstand the challenging environmental conditions. We present experimental evaluation of detection in various grasping scenarios such as grasping at an angle, corner grasping, improper closure of the gripper and proper grasp for logs of three different sizes. Results indicate the ability to detect and differentiate between good and poor grasping configurations.

## 1. Introduction

The new industrial revolution (*Industry 4.0*) is characterized by automation of traditional industrial processes, using modern technologies and communication networks. This is enabled by the use of smart sensors, edge computing, Artificial Intelligence (AI), and approaches for energy conservation and harvesting; leading to improved automation, self monitoring, failure prevention, and safer human–machine interaction [1,2].

Industrial robots are designed to automatically execute dangerous, exhausting and monotonous tasks and their successful automation mainly relies on the integration of advanced control and smart sensor systems. Sensors are responsible for providing the perception capabilities that robotic systems need to accomplish their tasks in an autonomous way, enabling a safe human–robot cooperation in the work environment. Visual, proximity and force sensors are commonly used in industrial robotics and especially in high throughput agroforestry applications [3,4,5,6]. Vision-based sensing combined with advanced control algorithms have been widely used in mobile applications for localization, mapping, obstacle avoidance, object recognition and manipulation [7,8]. Nevertheless, vision alone is a limited capability when the line of sight is obstructed or the illumination is inadequate, especially in mobile robotic systems with unforeseen surrounding conditions [9]. In particular, the grasping problem requires tactile sensing in addition to vision, as well as the integration of different sensing capabilities, as proposed in [10]. Visual–tactile information has been fused to achieve a stable grasping by improving the grasp prediction [11] and slip detection [12] methods using AI.

Previous research activities have focused on giving robots a human sense of touch, which is why they are usually equipped with tactile sensors [13,14] that provide robotic devices with learning and manipulation capabilities through highly sensitive fingers [15,16] or electronic skin [17,18]. As stated in [19], tactile intelligence is the future of robotic grasping and capacitive sensors have shown great advantages in this context, since they can be employed for both proximity and touch perception [20]. In [21], a multimodal capacitive sensor for typical manipulation tasks is presented. Static and dynamic sensing are integrated in the same layer of the capacitive sensor, which enhances sensitivity through direct written microstructured dielectric. A vision-based optical tactile sensor that measures geometry with high spatial resolution is addressed in [22], while an optoelectronics based tactile sensor capable of measuring force and torsional moment is presented in [23]. The classical robotic gripper with parallel fingers has been studied in [21,22,23]. However, the structural design requirements of a forestry crane gripper does not allow the direct application of the mentioned tactile sensing solutions. Logs are heavy objects and require curved grasping end-effectors to support the heavy weight. Moreover, the end-effector sensing capabilities of robotic arm grippers are improved with capacitive proximity sensors for contactless material detection and contour following assignments in [24] and [25], respectively. The capacitive sensing principle has also emerged as a promising technology for safety oriented applications. To guarantee a reliable Human Robot Interaction (HRI) in a safe working place, a Time-of-Flight (ToF) and self-capacitance sensors for human cooperative robots are combined in [26], while a capacitive skin is used for obstacle avoidance in [27].

### 1.1. Motivation and Related Work

Forestry cranes used for log manipulation present several challenges to automation as they are typically operated in uncontrolled outdoor environment. The constantly changing lighting conditions due to time of the day; change in weather conditions such as rain and snow; texture of the environment; and variation in the texture, size and shape of logs leads to difficulty in the use of camera based sensors. The presence of nearby unfelled trees, loading trucks and other equipment often restrict the workspace and limit the trajectories that the arms of the crane can take.

As a result, the use of a forestry crane still relies heavily on highly skilled human operators for complete operation. Manual control of cranes is both mentally and physically exhausting task, where the operator needs to spend several hours performing complex coordination while being subjected to harmful whole body vibrations [28]. Typically, crane operators spend about 80% of their time on controlling the crane [29] and on average are capable of only using 20% of the maximum velocity [28].

The potential economic benefits, as well as the alleviation of the human operator’s workload are some of the advantages that the automation of heavy-duty robots bring to the forestry industry [30]. Complete or partial automation of the crane can improve the efficiency and can reduce cognitive load on the human leading to easier operation and lesser training requirements, thereby bringing cost savings.

Hydraulic manipulators, similar to the one targeted in this work, have been recently investigated in [31,32,33]. In [28], the authors have equipped each joint of a commercial forestry machine with quadrature encoders to gather real time motion data that can be used to analyze crane motion patterns during manual control tasks driven by forestry operators. Additionally, a trajectory planning and motion control methods for a forestry harvesting crane have been presented in [34], where the control loop includes the data from encoders and pressure sensors installed on the piston joints and cylinders, respectively. Automatic log recognition approaches for forestry robots have been investigated in [35,36], both based solely on vision.

Research in the space of forestry crane automation has primarily focused on motion planning of the crane’s links and their positioning [37], and very little attention has been given to the automation of the end effector (gripper).

The system targeted in this work is a 5-Degrees-of-Freedom (DoF) hydraulically actuated crane, shown in Figure 1. It is intended to manipulate logs in forestry sawmills, e.g., in automatic placement of unordered (dumped) logs on conveyor belts for further processing. The investigations leading to our results have been performed on a 1:5 scaled version of the real crane [38], displayed in Figure 1b. The crane model is equipped with Inertial Measurement Units (IMUs), draw wire sensors, monochrome cameras, hydraulic pressure sensors and angular position sensors [39]. In addition, the workspace of the crane model is captured by depth and OptiTrack cameras installed inside the protecting cage. Nevertheless, none of those sensors is suited for being installed into the gripper, either due to an inadequate sensing principle or the inability to withstand strong impacts.

Despite the amount of studies that have focused on the automation of industrial robots, the advantages offered by capacitive sensors for improving grasping in forestry robots have not been sufficiently exploited. Capacitive sensors can overcome the difficulty of including other sensors into the end-effector claw. Preliminary results have been presented in [40] with the design of 3D- and inkjet-printed capacitive sensors intended to be included into the grippers of the hydraulic forestry crane under investigation. The sensors were integrated into a gripper prototype and evaluated in an experimental setup, showing sensitivity values in the fF order. However, the integration into the scaled crane model introduces new challenges, as higher noise levels and rough claw motions, and consequently a higher sensitivity is required. This is why a novel design of the sensing system is presented in this work to fulfill the conditions imposed by the hydraulic crane model. The proposed Smart Transducer Interface Module (STIM) not only offers a new design of the sensing elements, with different geometry, materials and fabrication procedure compared to the solution in [40], but also an upgraded measurement hardware, capable to establish communication with the crane controller computer in compliance with the Transducer Electronic Datasheet (TED) standard.

### 1.2. Contribution

In this paper, we design and develop a capacitive smart transducer for crane’s gripper, which can be used to safely automate the grasping mechanism of felled logs. The designed sensing system is suitable for retrofitting into existing industrial cranes with minimum modification. It consists of single ended electrodes placed on the claws of the grippers and the capacitance of the electrodes are measured using STIM. The STIM consists Capacitance to Digital Converter (CDC) connected to a microcontroller and energy harvesting circuit all contained in a compact Printed Circuit Board (PCB), which can be mounted within the gripper.

The STIM is especially designed for, but not restricted to, the gripper of the industrial forestry robot shown in Figure 1, in which the gripper is exposed to harsh environmental conditions, such as strong friction forces, mechanical impacts, dirt, changing climate characteristics and wide temperature ranges. In addition to its endurance to a hostile environment, capacitive sensors benefit from their simplicity, easy implementation, fast and inexpensive manufacturing process, as well as multi-modality and low latency data processing.

The proposed smart transducer provides robust information that can support the grasping process and improve the log processing speed. The aim is to employ tactile sensing that complements the vision based and pressure sensors, supplying additional information about the quality of the grasp. Our smart transducer is powered by an energy harvesting system using solar cells and batteries and can be safely placed within a housing cavity in gripper claws. The STIM transmits the measured sensor capacitances to the crane controller according to IEEE 1451.0 TEDs specification via Bluetooth Low Energy (BLE).

The system is evaluated in terms of the change in capacitance experienced in different grasping scenarios. The grasping behavior is characterized with logs of three different sizes during several grasping experiments performed using the manual control of the scaled crane model. The experimental results show that the STIM can be applied to a hydraulic forestry robot, providing useful data about the quality of the grasp during the log manipulation process. The capacitive sensing principle allows taking one step more toward the full automation of hydraulic forestry cranes operating in tough industrial scenarios.

## 2. System Description

Forestry cranes are commonly used in sawmills to handle logs for further processing, but also as truck mounted systems for portable applications. Usually, the cranes are manually controlled by a human operator. To automate the grasping operation, accurate feedback of the grasping process is needed by the automation controller. Optimal placement of sensors and data acquisition electronics within the gripper can provide the necessary information about the grasp status.

### 2.1. Sensing Principle

The proposed sensing mechanism consists of multiple single-ended capacitive sensors. In the single-ended principle an excitation source is connected to the electrode (transmitter) generating an electric field as illustrated in Figure 2. To measure single-ended capacitance, the electrode is excited and the displacement current is measured. When an object, in this case a log, comes close to the electrode it causes an electric field distortion, which is proportional to the change in the displacement current and consequently a change in the capacitance. The single-ended principle could also be seen as a virtual capacitor, which is formed when the object comes close to the sensor, with the electrode acting as one plate and the object acting as the other plate of the virtual capacitor. To guarantee a correct sensor operation, active shielding is used. It helps to eliminate offset capacitance introduced by the measurement cables.

In single-ended configuration, the capacitance between the excited electrode and the open environment or distant ground is measured, hence it is commonly used for proximity and touch perception, needed for the target gripper. The capacitance of the electrode is affected by the presence of nearby objects, their dielectric properties and distance from the sensor. The sensors are also susceptible to changes in ambient environmental parameters such as pressure, humidity, temperature, etc. (capacitance drift). However, in the proposed application, the change in capacitance due to log position is much faster than the capacitance drift, hence the effects of drift can be ignored and the drift can be regularly compensated with calibration measurements.

### 2.2. Gripper Design and Sensor Placement

The gripper consists of two parts (claws) that rotate in opposite directions when closing. Each claw comprises two fingers connected together by mid-plates as shown in Figure 3. The gripper is designed in such a way that the inner claw moves within the clearance between the fingers of the outer claw during the closing motion of the gripper. The claws are designed with a cavity that can hold a PCB inside them for housing the sensor electronics.

Two sensor arrays are placed each on the left and right finger of the outer claw. An expanded view of the sensor array is shown in Figure 3. The sensor array consists of three single ended capacitance electrodes (top, middle and bottom). The electrodes are placed on a layer of active shield separated by an insulation layer. The sensors and active shield layer are fully encapsulated by insulation layers and placed firmly on the fingers as illustrated in Figure 3. The active shield layer and the sensing electrodes are constructed using copper strips, while the insulation layers are performed with polyvinyl chloride strips, commonly known as isolation tape. The length of bottom and middle electrode strips are 6 cm and the length of the top electrode is 5 cm. The arc length of the fingers in the outer claw is roughly 17 cm and during grasping, the bottom and middle electrodes are more in direct contact with the log than the top electrodes due to the physical structure of the gripper, which is why they are slightly longer. All electrodes had a width of 1 cm, thickness of 0.06 mm and a separation of 2 mm is maintained between consecutive electrodes. The sensor electrodes are connected to the measurement circuit using coaxial cables whose outer conductor are connected to active shield. The electrodes’ capacitance on the left finger are denoted as C_TL_, C_ML_ and C_BL_ (top, middle and bottom). Similarly, for the electrodes on the right finger the capacitances are denoted as C_TR_, C_MR_ and C_BR_. The electrodes’ assembly, connection to the measurement hardware and installation in the gripper have been performed manually; therefore, they are easy to reproduce and their fabrication process is fairly low cost.

### 2.3. System Overview

A simplified block diagram of the complete system is shown in Figure 4. The crane hardware includes integrated sensors, actuators and an embedded controller.

The algorithm for controlling the crane position and grasping (crane controller) is implemented in PC. This high level controller can operate the crane in automatic [41] or manual (joystick) control mode. The controller computes the required forward kinematic parameters for the desired joint states and generates the suitable control commands for the crane’s embedded controller. The gripper’s hydraulic system has a pressure sensor whose output is used by the crane controller to prevent over-actuation of the gripper hydraulics. During grasping, the high level crane controller continuously monitors the hydraulic pressure sensor output. When the log is firmly grasped, the output of the force sensor increases when the hydraulics are actuated. If this output exceeds a predefined threshold, the high level crane controller ignores further commands from the control hardware (joystick) to prevent damage to the logs.

#### 2.3.1. Design and Realization of STIM

The STIM for grasp sensing is placed in the cavity of the gripper finger and connected to the sensors using shielded cables. The sensing system consists of six single-ended capacitance electrodes placed in the claw of the crane, as detailed in Section 2.2. The electrode capacitances are measured by AD7147 [42] CDC. The CDC is configured to measure each electrode in a time division multiplexed manner in six stages. The CDC is connected to a NRF52832 [43] micro-controller via I^2^C interface. The micro-controller and CDC are powered by an energy harvesting system using solar panels and a battery backup. For energy harvesting and battery management, BQ25570 [44] energy harvesting IC is used and configured to provide 3.0 V output. Figure 5a illustrates the complete sensing system. The entire circuit is deigned and fabricated on PCB of a suitable form factor that can fit in the cavity of the crane’s claw, as shown in Figure 5b.

#### 2.3.2. Wireless Network

The NCAP framework is a program that is capable of communicating with IEEE 1451.0 compliant smart transducers [45]. The framework can communicate with the devices using different communication interfaces such as BLE, Near-field Communication (NFC), ROS, etc. The framework initializes the data acquisition by first reading the TEDs from the STIM in a one-time setup process for the logical sensor integration and calibration into the system. After optional encryption of the communication, the measurements are streamed to the NCAP base station. During measurement, the framework continuously receives data from the STIM and decodes the received data according to the TEDs specifications, where the values for each sensor are checked to verify if they are within the specified bounds. The decoded data is then published to corresponding ROS topics as required and additionally saved to a measurement log file for future use. The system, addressed in [46], helps to speed up the logical integration of new devices and handles the communication between the sensor and the crane control computer. Moreover, additional software modules and drivers are not required, reducing the risk of errors and incompatibilities.

The minimum latency of the BLE link could be configured through the connection interval to be even higher in order to decrease the power consumption of the wireless link. For the proposed application and considering the speeds of the gripper hydraulics, this latency is acceptable.

#### 2.3.3. Grasp Detection Algorithm

To demonstrate proof of concept a simple grasp detection logic based on capacitance threshold crossing is implemented. The goal of the grasp detection algorithm is to identify a good grasp based on the information provided by the capacitive sensors. The output is set to logic high when a proper firm grasp is detected, otherwise the output will be logic low. During the initial calibration phase, the sensor parameters such as offset and maximum value were experimentally determined. For the grasp prediction, the measured electrode capacitance are corrected for offset and normalized. The normalized values of each electrodes are compared to the set threshold limit to obtain grasp detection for each electrode. For each electrode pair, the grasp is classified as good, poor or not detected. If grasp is detected in both left and right electrodes of an electrode-pair, this is indicative of a good grasp. Similarly, if a grasp is detected in only one of the electrodes in an electrode pair, this is indicative of a poor grasp. In the final step, the output is computed based on the grasp detection classification of each electrode pair. The output is logic low when all electrode pairs classify grasp as not detected, and when poor grasp is detected by any of the electrode pairs the output is logic low. The output is logic high otherwise. A pseudocode for the algorithm is given in Algorithm 1.
**Algorithm 1:** Grasp Detection Algorithm
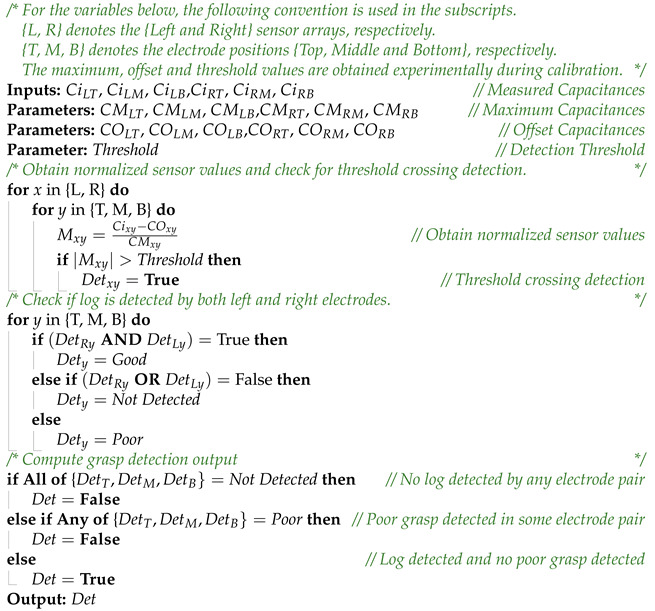


## 3. Experimental Results

In order to evaluate the proposed approach, a prototype of the STIM is developed and installed on the scaled down model of an industrial crane. Figure 6 shows the construction and placement of the sensors and the smart transducer interface. The sensors were installed on the outer claw of the crane on both the left and right fingers.

The STIM is designed and fabricated on PCB in a shape suitable to fit in the housing cavity provided within the finger body. The PCB is 1.6 mm thick and has four layers. A picture of the fully assembled PCB has been shown in Figure 5b. The PCB was mounted within the housing cavity on an insulating layer and connected to the electrodes, battery and solar cells. The energy harvesting system in the PCB harvests from ambient light using solar cells and a coin cell battery for energy storage.

Using a source measurement unit [47], the power consumption of the PCB was measured with the solar cell disconnected. The NRF52 microcontroller and the CDC are operating in full power mode and maximum transmission rate. The PCB consumes on average about 12 mW of power during the operation. This power consumption can be further reduced by operating the NRF52 microcontroller and the CDC in low power mode and by reducing the measurement and BLE transmission rate.

### 3.1. Grasping Experiments

To measure the performance of the sensors under various grasping scenarios, experiments were performed on logs of three different dimensions and under different grasping conditions, using the crane’s manual control mode. The diameter of the logs were 13.5 cm (big log), 7.5 cm (medium log) and 2.5 cm (small log). Figure 7 shows the logs that were used in the grasping experiments.

Sensor capacitance was measured under the following grasping conditions.

Proper grasp: The log is firmly grasped by the claws, the grippers are fully closed and the sensors are under tight contact with the log. When the crane is moving, the log does not experience any shift in its grasped position.Corner grasp: The log is held by the claws near its end, and is only partially held by the claw.Incomplete grasp: The claws are not closed to the maximum possible extent, hence the log is not held firmly by the claw.

### 3.2. Grasp Detection Results

Initially, the grasp sensing was tested for big logs under different grasping scenarios. Figure 8 shows the results obtained. First, proper firm grasp of the log was tested repeatedly at various speeds. The observed variation in electrode capacitance is shown in Figure 8a, here we can see that the capacitance of electrodes in both the left and right claws increase significantly and the change in capacitance is consistent over repeated grasping motions. The profile of the capacitance change depends on the speed of grasp/release as well as the duration of the grasp.

Subsequently, the performance of the sensor under corner grasping scenario (for big log) was tested, the results of which are shown in Figure 8b. In this experiment, only the right finger of the claw was in contact with the claw during the grasp. Here we can observe that only the capacitance of the sensors in the right finger is affected during the grasp, as was expected. Another variation of improper grasp that can often occur for big logs is grasping at an angle. Such a grasp can occur when picking up a log that is on top of another at an inclination. Experimental results obtained for grasping at an inclination are shown in Figure 8c. During the grasp, the log was mostly held by the claw’s right finger and touching the bottom electrode. Therefore, there can be seen a change in C_BR_ during the grasp.

The ability of the sensor to detect grasp firmness (for big logs) during crane movement was tested next. Initially an incomplete grasp was performed on the log and the crane was moved (including fast jerky motions) over the work space. The results are show in Figure 8e. Initially (t < 150 s), we can see that the sensors’ capacitances are not stable during the crane movement and often experience large change in value. At t = 150 s, firm grasp is performed and the crane is moved again. Here we can see that the sensor capacitance remains stable, indicating firm proper grasp.

Fingertip grasping can occur in automated grasping relying on top down camera images. In this scenario the log is grasped by the tips of the gripper fingers and not fully supported by the claws. The gripper has two hydraulic actuators, one for closing and the other for opening the gripper. The pressure of the hydraulic fluid in each actuator is measured by two independent pressure sensors. Figure 8f shows the output of the crane’s built in pressure sensors in the hydraulic system and the output of the proposed sensors. During the grasp the hydraulics system senses an increase in pressure and signals grasp completion. However, the proposed sensors did not detect the presence of a log within the gripper and this information can be used to identify the fingertip grasp scenario.

The ability of the sensors in grasp detection of medium and small sized logs was tested next, the results obtained are shown in Figure 9. Experimental validation of proper grasp detection for medium log was performed first and the results are plotted in Figure 9a. When compared to proper grasp of the big log, with the medium sized log the middle electrodes are much more sensitive to the presence of the log. This is due to the geometry of the crane claw and the difference in log sizes. When the big log is fully grasped, it is mainly in contact with the bottom sensors. Similarly when the medium is grasped, the middle sensors are in contact with the log.

The performance of the sensor in detecting corner grasp of the medium sized log was tested next. In this experiment, the crane lifted the log only using the right fingers. As expected, we can see in Figure 9b that only sensors in the right finger are responsive to grasp, while the left finger sensors are unaffected.

Afterwards, the performance of the sensor in detecting incomplete grasp was tested. In this experiment, initially a near complete grasp was performed ( 5 s < t < 45 s). The crane was moved around over the work-space and as we can see in Figure 9c, the sensor output experiences instability during the motion. Then, the crane was fully closed to perform a proper grasp ( 45 s < t < 115 s). Under this scenario, the sensor output was stable even during jerky motions of the crane, indicating a proper grasp. After this, the claws were partially opened to perform a loose grasp ( 115 s < t < 180 s). Under this condition the sensor output is much more prone to variation due to the crane movement.

Finally, the ability of the sensor to detect grasp of the small log was tested. The results are shown in Figure 9e. As the grasp progresses, the log first comes in contact with the bottom sensors (t = 2.5 s), and the sensors’ capacitances increase. As the log rolls over from the bottom sensors to the middle sensors (t = 5.5 s), the middle sensors increase in capacitance and the output of the bottom sensors decreases. Finally the log rolls over to the top sensors (t = 9.5 s), and the capacitance of the top sensors increase. In this configuration, the claws are fully closed, and the grasp is complete. At (t = 16 s), the claw is opened slowly and the log is released.

## 4. Conclusions

In this paper, we present a capacitive smart transducer for grasp detection application in a forestry crane robot. The sensing system consists of six single-ended electrodes placed on the fingers of a gripper claw, whose capacitance are measured using IEEE 1451.0 compliant STIM. The STIM is a low power unit that can be powered using solar energy harvesting system with battery backup. Experimental evaluation of the grasp detection capability was performed on logs of three different dimensions and under different grasping scenarios on a scaled down replica of an industrial crane. Results indicate the suitability of the proposed design to detect successful firm grasp for different sized logs. 

## Figures and Tables

**Figure 1 sensors-23-02747-f001:**
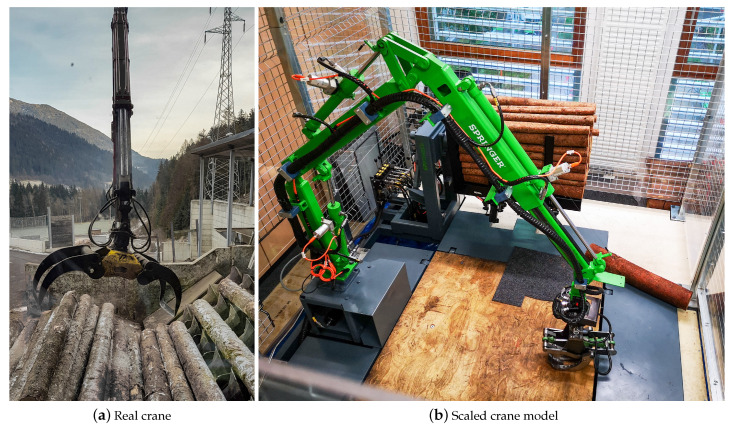
Hydraulic forestry crane for autonomous log manipulation. (**a**) Real crane, picture of a mobile forestry crane at a sawmill delivery and (**b**) scaled crane model, the 1:5 scaled model with hydraulics designed to match the scale and manual control equal to the real version. The scaled crane model has an arm length of 2.5 m and it is surrounded by a protective cage, providing a workspace volume of approximately 20 m^3^.

**Figure 2 sensors-23-02747-f002:**
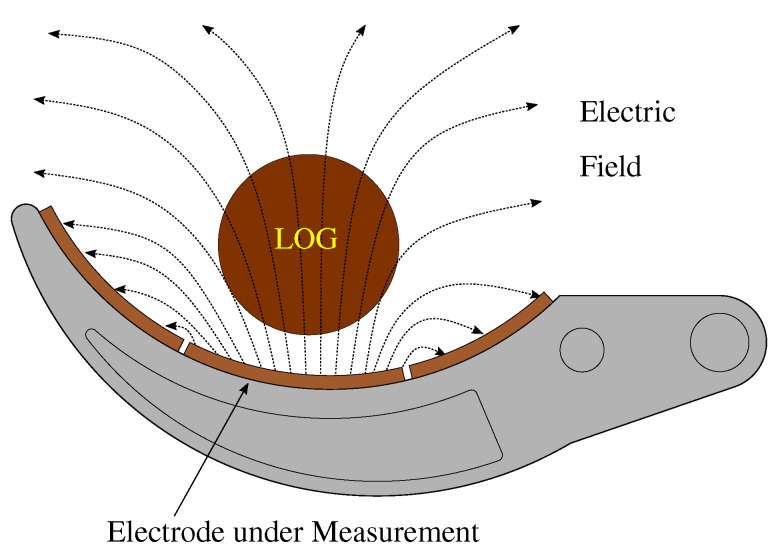
Single ended capacitance measurement sensing principle. The capacitance between the transmitter electrode and the environment is affected in presence of nearby materials and their dielectric properties.

**Figure 3 sensors-23-02747-f003:**
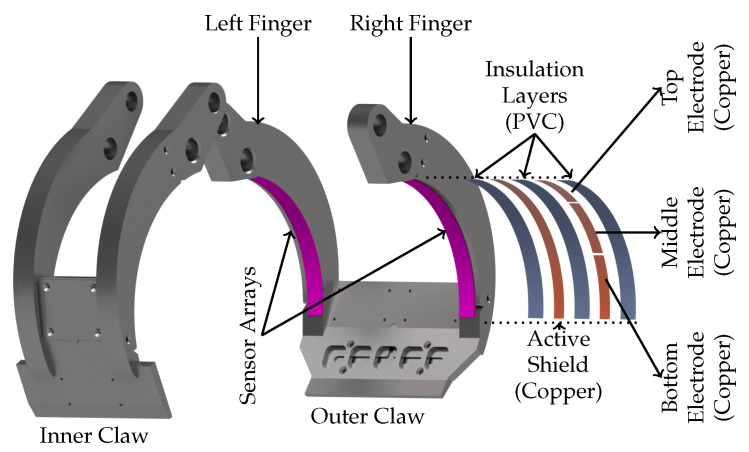
Design of gripper claw and sensors’ Placement. A sensors array consists of three transmitter electrodes (top, middle and bottom) as illustrated in the expanded view. The electrodes are placed over an active shield layer separated by an insulating layer. The electrode/shield layers are then fully enclosed in an insulating enclosure. The sensor arrays are placed each on both the left and right fingers of the the outer gripper claw.

**Figure 4 sensors-23-02747-f004:**
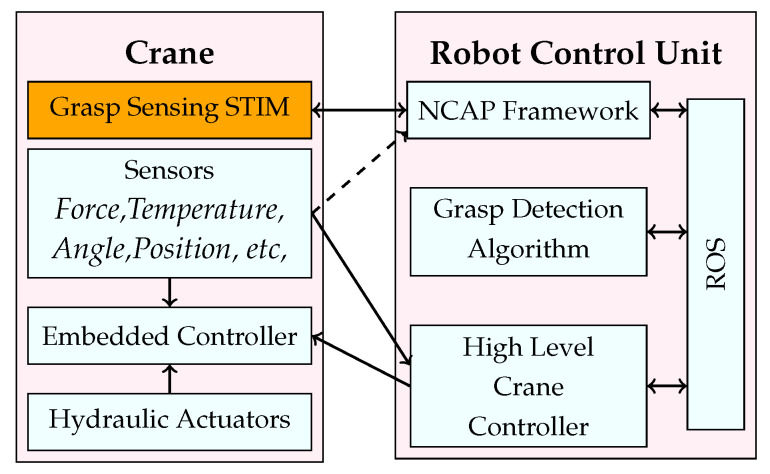
High level data acquisition and processing framework. The crane consists of various sensors for monitoring and hydraulic system for actuating the position control. It also has an embedded controller for controlling the crane based on external from the high level controller. The designed grasp sensing system (STIM) is mounted on the crane in a protective cavity. The STIM communicates to the Network Capable Application Processor (NCAP) framework in the host computer via BLE; the obtained data is then used for further processing and passed to a program using Robot Operating System (ROS).

**Figure 5 sensors-23-02747-f005:**
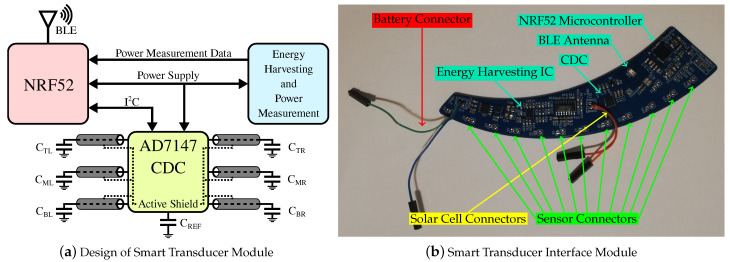
Sensor data acquisition system. Design of STIM is shown in (**a**). The power management harvests energy using solar panels and uses a coin cell as battery backup. NRF52 is used to configure and acquire data from the CDC, which is then transmitted via BLE in compliance with IEEE 1451.0 specifications to the crane controller. (**b**) The fully assembled PCB of the STIM and highlights the various important components related to capacitance measurement, energy harvesting and wireless transmission.

**Figure 6 sensors-23-02747-f006:**
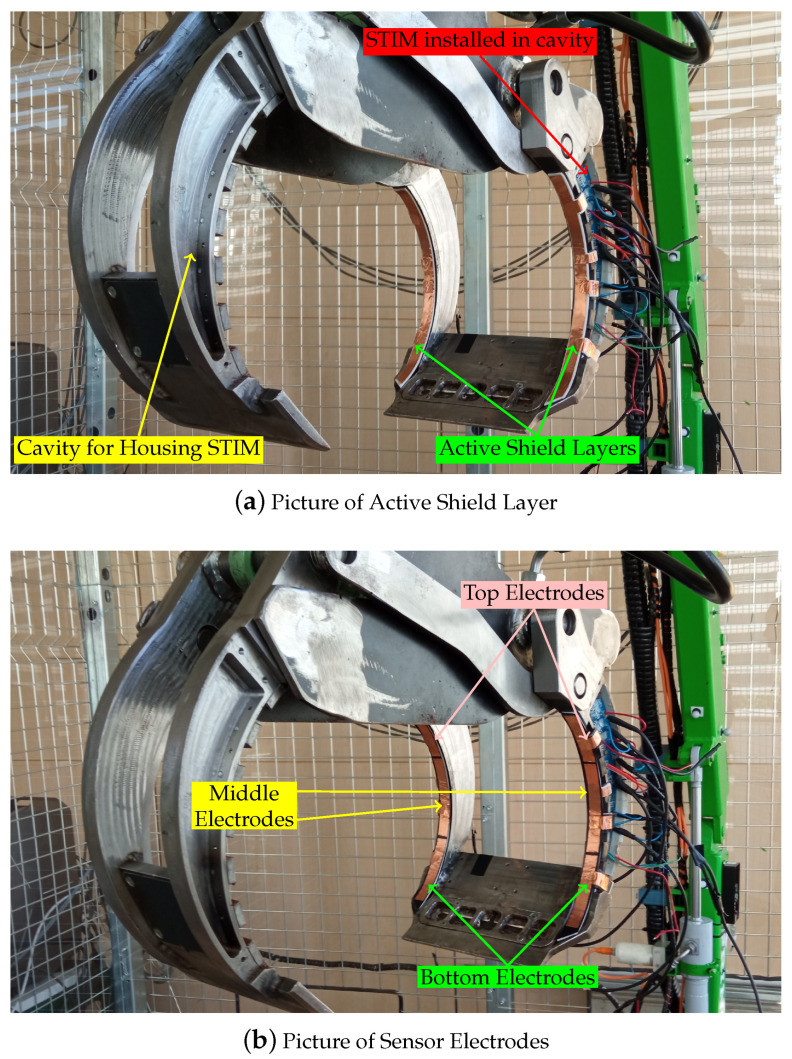
Experimental setup. (**a**) The placement of active shield layer on the gripper’s claw is shown. One of the cavities for holding the STIM is shown clearly on the gripper’s inner claw. The STIM is placed in a cavity on the right finger of the outer gripper. (**b**) The construction of the sensor electrodes.

**Figure 7 sensors-23-02747-f007:**
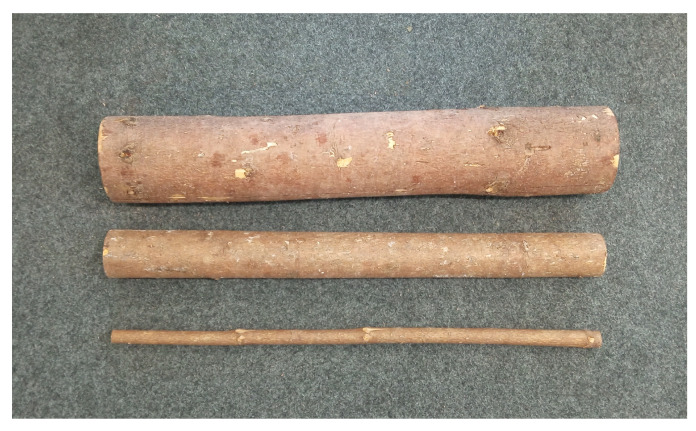
Logs of different diameters used in the grasping experiments. Big log (top) has a diameter of 13.5 cm, medium log (middle) has a diameter of 7.5 cm and small log (bottom) has a diameter of 2.5 cm.

**Figure 8 sensors-23-02747-f008:**
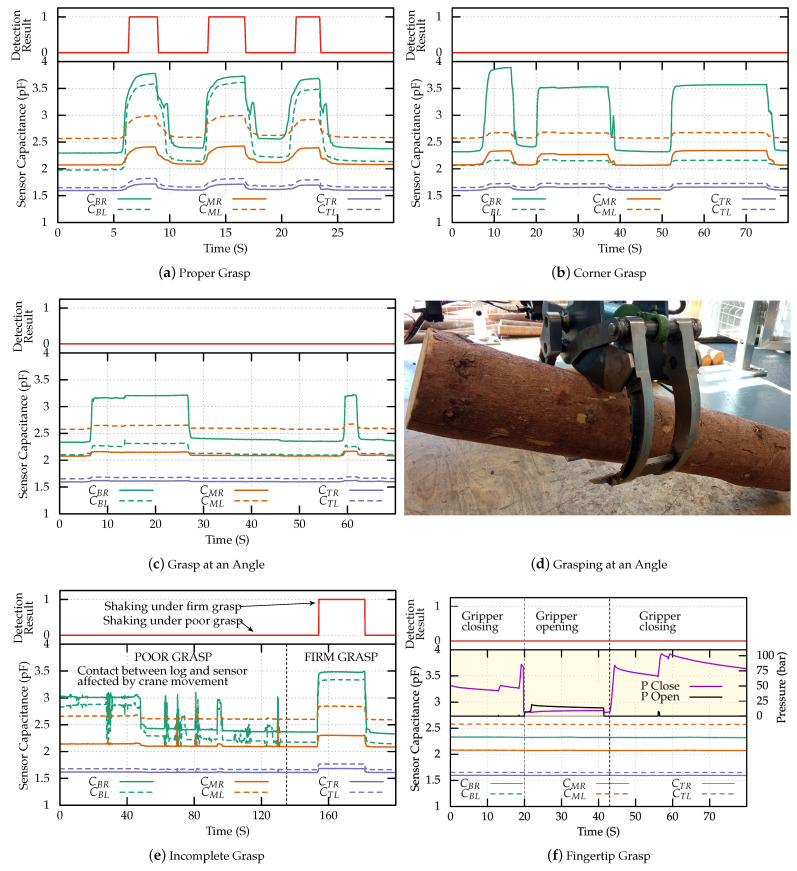
Experimental results for grasping big log (13.5 cm). The solid lines represent the electrodes on the right claw and dashed lines represent the electrodes on the left claw. The result of detection algorithm is plotted in red on top of each plot. (**a**) shows the output of the sensors during a proper firm grasp, (**b**) shows the output of the sensors when grasping the log in the corner, (**c**) shows the output of the sensors when grasping the log at an angle and a picture of angle grasping scenario is shown in (**d**), (**e**) shows the output of the sensors when moving the log under incomplete grasp and (**f**) shows the output of both the open and close hydraulic pressure sensors (right y-axis, yellow background); and the capacitive sensors (left y-axis) during fingertip grasp.

**Figure 9 sensors-23-02747-f009:**
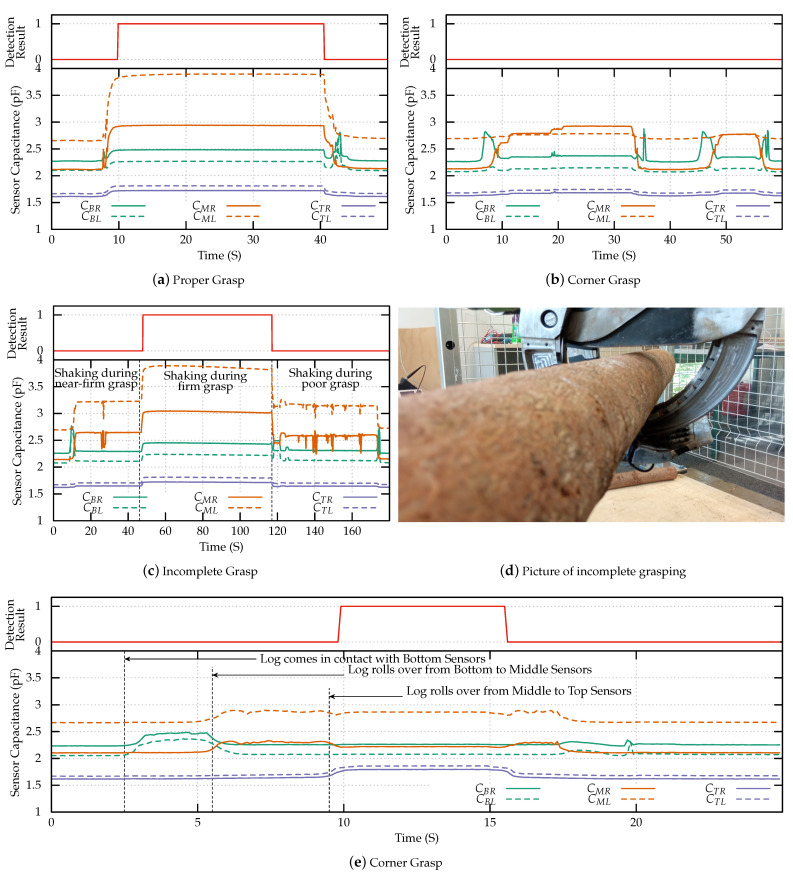
Experimental results for grasping medium (7.5 cm) log. The solid lines represent the electrodes on the right claw and dashed lines represent the electrodes on the left claw. The result of detection algorithm is plotted in red on top of each plot. (**a**) shows the output of the sensors during a proper firm grasp, (**b**) shows the output of the sensors when grasping the log in the corner and (**c**) shows the output of the sensors when moving the log under incomplete grasp, (**d**) shows a picture of incomplete grasping procedure and (**e**) shows the results of grasp detection for small log.

## Data Availability

Not applicable.

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
