# Peer review of "Design and Evaluation of Capacitive Smart Transducer for a Forestry Crane Gripper"

_sensors, 2023, doi:10.3390/s23052747_

Round 1

Reviewer 1 Report

The paper presents a capacitive smart transducer for grasp detection application in a forestry crane robot. The paper can be very interesting for readers. Results show the practical use of the presented approach of the capacitive smart transducer.

Author Response

Thanks to the reviewer for the positive feedback and for appreciating our research outcome. We also think that the presented Smart Transducer Interface Module (STIM) based on a capacitive sensing principle can bring the readers a fresh insight into the design and evaluation of wireless sensors for harsh environments and industrial applications.

Reviewer 2 Report

This paper presents a tactile sensor for the gripper claws of a forestry crane. The sensor is completely wireless and is powered using energy harvesting. The paper shows how the proposed sensor configuration is able to detect different grasping configurations.

The idea behind the paper is interesting, but the paper lacks some details, and some claims need to be justified. More in detail:

The wireless interface of the sensor seems an important feature highlighted even in the abstract but there is no actual evaluation of such an interface. What about latency, jitter, etc.? Such data are useful when using this sensor with a closed-loop controller. The same concern applies to energy harvesting, what is the power consumption of the sensor?

The related works miss a comparison of the described sensor with tactile sensors used in the more classical parallel robotics grippers (an example is the [R1] also published in MDPI Sensors).

It is not clear how the Robot Control Unit in Fig.3 uses the sensor data. This concern could be related to the Detection Result plots in the experiments. It is not clear how this signal is generated. These aspects should be described more in detail.

Minor comment: The legend in the experiment figures is hard to read, the labels cover some lines and it is not clear at a first look which lines are associated with which signal.

[R1] M. Costanzo, et al., “Design and Calibration of a Force/Tactile Sensor for Dexterous Manipulation,” Sensors, vol. 19, no. 4, p. 966, Feb. 2019, doi: 10.3390/s19040966.

Author Response

Dear Reviewer,

We would like to express our appreciation for all the insightful remarks and suggestions. Definitively, they helped to improve the quality of the manuscript and we sincerely hope that this revised version satisfies the requirements of the MDPI Sensors Journal.

Please find in the pdf file provided, the authors’ answers to the comments and questions of the reviewer about the manuscript, “Design and Evaluation of Capacitive Smart Transducer for a Forestry Crane Gripper”. In this response letter, the comments from the reviewers are written in black, while our answers are in red. Similarly, the modifications in the revised manuscript are also highlighted in red. The numbered references and figures correspond to the revised manuscript.  

Sincerely,

The Authors.

Reviewer 3 Report

1. The notes in Figure 1 should preferably provide information on the main dimensions of the crane scale model or the approximate volume of the working area of the protective cage.

2. In the second section, whether the insulating layer and capacitance sensor materials can be separated to illustrate.

3. In 2.2, the principle of capacitor sensor is not clearly stated.

4. It is mentioned in the beginning of 3.2 that Figure 5 is the experimental result of large diameter logs, but the diameter of logs in Figure 5 (e) and (g) seems to be very different.

5. The diameter of the captured log in Figure 6 (d) is 2.5cm. Is it possible? Is there a problem with the unit (bottom electrode and middle electrode length 6 cm, upper electrode length 5 cm)?

Author Response

(The authors gave the same response as above.)

Reviewer 4 Report

This manuscript describes an interesting application. However, several issues must be addressed before merit publication:

1) A similar protopype was published in [39]. The contributions of this manuscript over that paper must be highlighted in the Introduction.

2) The Experimental Results Section is the most important section of the paper. It is illustrative of the sensing capabilities of the proposed sensor array, but these capabilities were already proven in [39]. Then the contribution is questionable. In order to increase the interest of the contribution, these sensing experiments must be completed with something else. For example, with a decision making algorithm that recognizes the different kind of grasps in real time from these measurements and takes basic decisions in order to improve grasping.

3) Are these the only sensors used in the grasping task?. If others were present, their role and measurements should be provided.     

Author Response

(The authors gave the same response as above.)

Round 2

Reviewer 2 Report

The authors answered all my points. I have no further concerns.

Just a very minor suggestion: To improve readability, consider better formatting Algorithm 1 (e.g., using a different style for code comments)

Author Response

Dear Reviewer,

In the revised paper the comments are formatted in C-style and colored dark green to improve the readability.

Thank you for your valuable time and efforts in reviewing the paper. The comments and suggestion were very helpful in improving the quality of the manuscript and hope that the revised manuscript satisfies the requirements of MDPI Sensors Journal.

Sincerely,

The Authors.

Reviewer 4 Report

The issues that I raised have been clarified. The article can be published.

Author Response

Dear Reviewer,

Thank you for your valuable time and efforts in reviewing the paper. The comments and suggestion were very helpful in improving the quality of the manuscript and hope that the revised manuscript satisfies the requirements of MDPI Sensors Journal.

Sincerely,

The Authors.